# A Selective Melatonin 2 Receptor Agonist, IIK7, Relieves Blue Light-Induced Corneal Damage by Modulating the Process of Autophagy and Apoptosis

**DOI:** 10.3390/ijms252011243

**Published:** 2024-10-19

**Authors:** Hyeon-Jeong Yoon, Enying Jiang, Jingting Liu, Hui Jin, Hee Su Yoon, Ji Suk Choi, Ja Young Moon, Kyung Chul Yoon

**Affiliations:** Department of Ophthalmology, Chonnam National University Medical School, and Hospital, Gwangju 61469, Republic of Korea; yoonhyeonjeong@daum.net (H.-J.Y.);

**Keywords:** melatonin type 2 receptor, IIK7, blue light, autophagy, apoptosis, corneal damage

## Abstract

This study aims to investigate the effect of the selective MT2 receptor agonist, IIK7, on corneal autophagy and apoptosis, aiming to reduce corneal epithelial damage and inflammation from blue light exposure in mice. Eight-week-old C57BL/6 mice were divided into BL-exposed (BL) and BL-exposed with IIK7 treatment (BL + IIK7 group). Mice underwent blue light exposure (410 nm, 100 J) twice daily with assessments at baseline and on days 3, 7, and 14. Corneal samples were analyzed for MT2 receptor expression, autophagy markers (LC3-II and p62), and apoptosis indicators (BAX expression and TUNEL assay). Then, mice were assigned to normal control, BL, and BL + IIK7. Ocular surface parameters, including corneal fluorescein staining scores, tear volume, and tear film break-up time, were evaluated on days 7 and 14. On day 14, reactive oxygen species (ROS) levels and CD4^+^ IFN-γ^+^ T cells percentages were measured. The BL group exhibited higher LC3-II and p62 expression, while the BL + IIK7 group showed reduced expression (*p* < 0.05). The TUNEL assay showed reduced apoptosis in the BL + IIK7 group compared to the BL group. ROS levels were lower in the BL + IIK7 group. The BL + IIK7 group showed improved ocular surface parameters, including decreased corneal fluorescein staining and increased tear volume. The percentages of CD4^+^ IFN-γ^+^ T cells indicated reduced inflammatory responses in the BL + IIK7 group. The MT2 receptor agonist IIK7 regulates corneal autophagy and apoptosis, reducing corneal epithelial damage and inflammation from blue light exposure.

## 1. Introduction

The rapid increase in the use of digital LED devices has significantly elevated blue light (BL) exposure, particularly within the 400–490 nm wavelength range [1]. Blue light penetrates deeply into the eye and potentially damages the cornea and other ocular tissues [2,3]. Studies have demonstrated that excessive exposure to BLs can induce oxidative stress, apoptosis, and autophagy dysfunction, all of which contribute to corneal damage and inflammation [2,4,5,6,7]. This damage is particularly concerning given the crucial role of the cornea in protecting the eye and maintaining visual clarity [2,6]. Therefore, developing therapeutic strategies to mitigate these adverse effects is essential.

Melatonin, a neurohormone primarily produced by the pineal gland, is known for its protective effects against oxidative stress and inflammation in various tissues, including the eyes [8,9,10,11]. Melatonin exerts these effects through its receptors, MT1, MT2, and MT3/QR2, which are distributed in ocular tissues such as the cornea and retina [12,13,14]. The MT2 receptor, a specific subtype of the melatonin receptor, plays a pivotal role in maintaining corneal health on the ocular surface [14,15]. The MT2 receptor is crucial for the regulation of cellular homeostasis and may be a target for protecting the cornea from BL-induced damage [14,15,16].

The activation of MT2 receptors has been shown to modulate autophagy and apoptosis, reducing oxidative stress and promoting cell survival in the corneal epithelium [9,15,17]. Our previous studies investigated the effects of BL exposure on the expression of melatonin and its receptors (MT1 and MT2) in the mouse cornea. Our research revealed that BL exposure can induce the local production of melatonin and upregulate MT2 receptor expression in the corneal epithelium [15]. Additionally, we found that MT2 receptor expression is closely associated with the regulation of apoptosis and impaired autophagy in response to BL exposure [15].

Therefore, this study aims to explore the therapeutic potential of IIK7 (N-butanoyl-2-[2-methoxy-6H-isoindolo(2,1-a)indol-11-yl]ethanamine), a selective MT2 receptor agonist, in protecting the cornea from BL-induced damage. IIK7 has been shown to improve tear secretion and accelerate corneal wound healing, highlighting its potential as a treatment for ocular surface diseases [12,14,18]. In this study, we investigated the effects of IIK7 on autophagy and apoptosis in a mouse model of BL-induced corneal injury, focusing on its ability to regulate these processes and reduce reactive oxygen species (ROS). Additionally, we aim to explore the protective mechanisms of IIK7 and its potential as a therapeutic agent for BL-induced ocular surface disorders by assessing ocular surface parameters, including tear production, corneal damage, and inflammation.

## 2. Results

### 2.1. MT2 Receptor Expression in Corneal Tissue

The mean MT2 expression in the normal control group was 41.68 ± 6.07 MFI. The MT2 expression in the BL group was 57.04 ± 4.44 MFI on day 3, 63.62 ± 9.19 MFI on day 7, and 80.77 ± 5.73 MFI on day 14 compared to a baseline of 41.68 ± 6.07 MFI. In contrast, the BL + IIK7 group showed MT2 expression levels of 71.18 ± 9.05 MFI at day 3, 66.52 ± 5.25 MFI at day 7, and 55.58 ± 2.46 MFI at day 14. Both the BL-exposed group and BL + IIK7 treatment group exhibited increased MT2 expression in the cornea on days 3 and 7 compared to baseline (all *p* < 0.01). Notably, the BL group showed significantly increased MT2 expression on day 14 compared with day 3 (*p* < 0.01). However, the BL + IIK7 group demonstrated a significant reduction in MT2 expression by day 14 compared with the BL group (*p* < 0.01), suggesting that IIK7 modulates the overexpression of MT2 receptors induced by prolonged BL exposure (Figure 1).

### 2.2. Autophagy Markers (LC3-II and p62)

Autophagy was evaluated by analyzing the expression of LC3-II and p62. The mean LC3-II level in the normal control group was 5.31 ± 0.12 MFI. LC3-II levels in the BL group were 45.55 ± 2.81 MFI on day 3, 42.22 ± 5.65 MFI on day 7, and 40.45 ± 4.76 MFI on day 14. In the BL + IIK7 group, LC3-II levels were 33.41 ± 3.49 MFI on day 3, 29.20 ± 1.13 MFI on day 7, and 23.65 ± 4.37 MFI on day 14. Both groups showed increased LC3-II expression on days 3, 7, and 14 compared to baseline, indicating an upregulation of autophagy in response to BL exposure (all *p* < 0.01). However, the BL group exhibited significantly higher LC3-II expression than the BL + IIK7 group at all time points (all *p* < 0.01; Figure 2A,C).

The mean p62 level, a marker of impaired autophagy, in the normal control group was 18.06 ± 6.48 MFI. The expression of p62 was 28.05 ± 3.86 MFI on day 3, 28.65 ± 3.69 MFI on day 7, and 39.47 ± 3.19 MFI on day 14 in the BL group. In the BL + IIK7 group, p62 expression was 24.70 ± 5.68 MFI on day 3, 26.38 ± 1.27 MFI on day 7, and 26.09 ± 4.57 MFI on day 14. The expression of p62 was significantly higher in the BL group on day 14 compared to baseline (*p* < 0.01). In contrast, the BL + IIK7 group showed markedly lower p62 expression (*p* = 0.04), indicating that IIK7 helps preserve the autophagic process by preventing impairment (Figure 2B,D).

### 2.3. Apoptosis Indicators (BAX Expression, TUNEL Assay)

The mean BAX expression level in the normal control group was 16.38 ± 2.57 MFI. The BL group demonstrated BAX expression levels of 25.99 ± 2.60 MFI on day 3, 25.24 ± 1.86 MFI on day 7, and 23.77 ± 6.85 MFI on day 14. The BL + IIK7 group showed BAX expression levels of 15.84 ± 3.25 MFI on day 3, 23.02 ± 1.77 MFI on day 7, and 20.09 ± 2.16 MFI on day 14. The BL group showed increased BAX expression on day 3, indicating higher levels of apoptosis (*p* = 0.04; Figure 3A,B). The TUNEL assay results further supported the anti-apoptotic effects of IIK7 with apoptotic cell counts in the BL group being 9.25 ± 2.38 cells/100 µm on day 3, 8.2 ± 2.04 cells/100 µm on day 7, and 7.4 ± 1.62 cells/100 µm on day 14 compared to significantly lower counts in the BL + IIK7 group at 1.6 ± 1.02 cells/100 µm on day 3, 3.2 ± 0.98 cells/100 µm on day 7, and 1.6 ± 0.49 cells/100 µm on day 14 (all *p* < 0.01, Figure 3C). The mean apoptotic cell count in the normal control group was 0.25 ± 0.43 cells/100 µm.

### 2.4. Reactive Oxygen Species (ROS) Levels

In the cornea, the BL group showed an ROS level of 232.75 ± 12.33% and the BL + IIK7 group showed an ROS level of 179.62 ± 12.61% compared to normal control. The ROS level in the conjunctiva was found to be 232.75 ± 12.33% in the BL group and 179.62 ± 12.61% in the BL + IIK7 group compared to the normal control group. The results showed that ROS levels were significantly lower in the BL + IIK7 group than in the BL group (*p* = 0.04, cornea; *p* = 0.01, conjunctiva; Figure 4).

### 2.5. Ocular Surface Parameters and Inflammatory T-Cell

The ocular surface was evaluated by corneal fluorescein staining (CFS), tear film parameters, and inflammatory cell infiltration. On day 3, the BL group had an average CFS of 8.20 ± 1.94, which increased to 9.66 ± 1.63 on day 7 and further to 11.05 ± 2.01 by day 14. In comparison, the BL + MT2 agonist group had a lower average score of 7.27 ± 1.69 on day 3, which increased slightly to 8.26 ± 1.44 by day 7 and to 9.27 ± 1.22 by day 14. The BL + IIK7 group exhibited significantly decreased corneal staining scores on days 7 and 14 compared with the BL group, indicating reduced corneal damage (all *p* < 0.01; Figure 5).

Tear volume on day 7 in the normal control group was 0.041 ± 0.005 µL, the BL group was 0.023 ± 0.004 µL, and the BL + IIK7 group was 0.025 ± 0.007 µL. On day 14, the tear volumes were 0.040 ± 0.005 µL in the normal control group, 0.018 ± 0.005 µL in the BL group, and 0.023 ± 0.006 µL in the BL + IIK7 group. The tear volume was significantly higher in the BL + IIK7 group on day 14 than in the BL group (*p* = 0.02). For tear break-up time (TBUT), on day 7, the normal control group had a TBUT of 2.8 ± 0.5 s, while the BL group reduced to 1.3 ± 0.4 s, and the BL + IIK7 group showed a TBUT of 1.6 ± 0.5 s. On day 14, the normal control group maintained a TBUT of 2.7 ± 0.4 s, the BL group showed 1.2 ± 0.3 s, and the BL + IIK7 group showed 1.4 ± 0.4 s. There were no significant differences in the TBUT between the groups (Figure 6).

The inflammatory response was assessed by measuring the percentages of CD4 + IFN-γ T cells in the cornea and conjunctiva. In the cornea, the percentage of CD4^+^ IFN-γ^+^ T cells was 26.00 ± 2.52 in the BL group compared to 4.32 ± 0.48 in the normal control and 12.66 ± 1.02 in the BL + IIK7 group. Similarly, in the conjunctiva, the percentage was 24.87 ± 1.26 in the BL group compared to 5.37 ± 0.86 in the normal control and 14.54 ± 1.50 in the BL + IIK7 group. In contrast, the BL + IIK7 group exhibited significantly lower percentages of CD4^+^ IFN-γ^+^ T cells compared to the BL exposure group (all *p* < 0.01; Figure 6C,D).

## 3. Discussion

BL, within the wavelength range of 400–490 nm, has been recognized as a contributor to ocular surface damage, which is a concern heightened by the increasing use of digital devices [1]. The cornea, the eye’s first line of defense, is vulnerable to BL-induced oxidative stress, inflammation, and cellular damage [2,4,5,6]. Recent studies have reported the role of melatonin and its receptors, particularly MT2, in the mitigation of these adverse effects. The MT2 receptor, expressed in corneal epithelial cells, has been shown to regulate cellular processes, such as autophagy and apoptosis, positioning it as a potential therapeutic target for protecting the cornea against BL-induced damage [12,14,18]. Due to its role in maintaining cellular homeostasis and reducing oxidative stress, the MT2 receptor may be a target for protective agents against the detrimental effects of BL exposure [12,14,18].

In our previous studies, we observed a time-dependent increase in MT2 expression in response to BL exposure [15]. Specifically, inhibiting MT2 with an antagonist led to increased impaired autophagy and apoptosis, which aggravated corneal epithelial damage [15]. The increase in MT2 expression over time reflects the need for cellular protection against the sustained oxidative stress caused by prolonged BL exposure. MT2 appears to play a central role in mitigating BL-induced damage, as MT1 did not exhibit upregulation in response to BL exposure, and the current literature provides no evidence for MT3 expression in the cornea [13,15]. Therefore, we focused on a selective MT2 agonist as the primary candidate for therapeutic intervention in this study.

IIK7 has high selectivity and affinity for the MT2 receptor, with a binding affinity 90-fold greater than for MT1, and it primarily acts through MT2 receptors [19]. It has been demonstrated that its action is inhibited by the MT2-specific antagonist DH97 [12,14]. Previous studies demonstrated that IIK7 enhances tear secretion and promotes corneal wound healing. The primary objective of this study was to evaluate the ability of IIK7, a selective MT2 receptor agonist, to protect the cornea from BL-induced damage [12,14,16,18]. Considering the known effects of MT2 on oxidative damage and inflammation, we hypothesized that IIK7 exerts a protective effect against BL-induced corneal surface damage.

Autophagy is a critical cellular process that maintains homeostasis by degrading and recycling damaged cellular components [20,21,22]. In this study, we observed that BL exposure led to the upregulation of autophagy markers such as LC3-II, indicating an initial attempt by corneal cells to manage BL-induced stress [23]. However, the increase in p62 levels in the BL group indicates impaired autophagy, leading to the accumulation of damaged cellular material [24]. In contrast, IIK7 treatment significantly reduced p62 levels compared to the BL group, demonstrating that IIK7 effectively modulates autophagy and reduces the accumulation of cellular damage in the corneal epithelium.

Autophagy plays a complex role under stress conditions [8,24,25]. While it generally serves as a protective mechanism by clearing cellular debris, excessive or dysregulated autophagy can lead to detrimental effects, as seen in diseases like dry eye disease [9,25,26,27]. In our study, the reduction in autophagy markers with IIK7 treatment suggests that it aids in restoring a balanced autophagic process, preventing excessive cellular damage and supporting corneal health. This highlights the dual role of autophagy, where proper modulation is the key to preventing pathological outcomes under prolonged stress, such as BL [9,26,27].

In this study, we measured apoptosis using the BAX and TUNEL assays. Apoptosis, or programmed cell death, is another crucial process affected by BL exposure, contributing to corneal epithelial damage [4,5,28]. The apoptosis observed in our study is primarily associated with the intrinsic (mitochondrial) pathway [29,30]. This pathway is regulated by BAX, a pro-apoptotic protein, which promotes mitochondrial outer membrane permeabilization, leading to the release of cytochrome c and the activation of caspase-9 and caspase-3 [29,30]. In both this and our previous studies, we observed that BL exposure upregulated both BAX and cleaved caspase-3, indicating the activation of the intrinsic pathway [15]. The significant reduction in TUNEL-positive cells in the IIK7-treated group indicated that IIK7 effectively mitigated BL-induced apoptosis, likely by enhancing protective pathways and reducing oxidative stress, preserving the integrity of the corneal epithelium. In our study, although BAX expression showed no substantial difference between the BL and BL + IIK7 groups, TUNEL analysis revealed a significant reduction in the number of apoptotic cells in the BL + IIK7 group. This discrepancy is likely due to the difference between the BAX protein, a pro-apoptotic factor that promotes apoptosis, and the TUNEL assay, which detects DNA fragmentation in the later stages of apoptosis [31,32].

As a result, IIK7 application significantly reduces ROS levels, helping to alleviate oxidative stress in the corneal epithelium caused by BL exposure. Mitochondrial dysfunction, a key trigger of the intrinsic apoptosis pathway, was associated with increased ROS production [33]. By stabilizing the mitochondrial function and lowering ROS levels, IIK7 mitigates oxidative stress, reducing both apoptosis and inflammation [33,34]. Moreover, by decreasing ROS, IIK7 indirectly suppresses inflammation by limiting the activation of inflammatory pathways like NF-κB and reducing pro-inflammatory cytokine production [32,34,35]. IIK7 also shows potential to improve ocular surface health by improving clinical parameters such as tear volume and CFS. This treatment also resulted in a significant reduction in inflammatory CD4^+^ IFN-γ^+^ T cells, highlighting its anti-inflammatory properties in alleviating BL-induced immune responses in the cornea.

Our study demonstrated the differential expression of MT2 receptors following BL exposure and IIK7 treatment. Although BL exposure led to an increase in MT2 expression in the cornea, administering IIK7 significantly reduced MT2 levels by day 14. Previous studies have shown that the upregulation of MT2 receptors may occur as a reactive change to respond to increased oxidative stress and cellular damage as a protective mechanism. Our findings suggest that IIK7 effectively regulates cellular stress responses by activating the MT2 receptor, reducing the need for sustained high levels of MT2 expression to cellular environment stabilizer.

The limitation of our study is that we focused on the acute effects of blue light exposure over a 14-day period. Extending exposure could lead to more complex and unpredictable physiological responses.

## 4. Materials and Methods

### 4.1. Mouse Models and Experimental Procedures

This study was approved by the Chonnam National University Medical School Research Institutional Animal Care and Use Committee (Approval No. CNUHIACUC-21056). All animals were treated in accordance with the ARVO Statement for Use in Ophthalmology and Vision Animal Research. The study employed female C57BL/6 mice, aged 7–8 weeks, and a temperature of 25 °C was maintained under 12 h contrast cycle illumination (bright: 08:00 a.m.–20:00 p.m.; dark: 20:00 p.m.–08:00 a.m.).

This study included a two-part experiment (Parts I and II) with light induction performed twice daily (irradiation began at 21:00 and 04:00 to avoid variation) for 14 consecutive days. In Part I, mice were exposed to BL at an energy dose of 100 J/cm^2^ (74 min) at 3, 7, or 14 days and then divided into two groups: BL-exposed (BL group) and BL-exposed with IIK7 treatment (BL + IIK7 group). In Part II, the mice were divided into three groups: normal control, BL exposed to 100J/cm^2^ (BL group), and BL exposed to IIK7 treatment (BL + IIK7 group). The mice in the normal control group were not exposed to BL during the experiment.

The BL groups were induced with the respective energy doses for 14 days, and 2 μL of MT2-selective agonist (IIK7, catalog no. ST110554; Timtec LLC, FL, USA) was administered to the BL + IIK7 group. IIK7 was dissolved in DMSO (30 mg/mL) and diluted with PBS to a concentration of 1 nM. The mice were confined to an adjustable retaining cage in a dark room, where the BL was placed 5 cm above and perpendicular to the head of the mouse. Only the ceiling of the cage emitted light. Therefore, the dose of light on the mouse ocular surface at any given instant depends on head posture. We estimated that on average, the mouse kept its head aligned with its body while in the retaining cage [7,15].

In Part I, on days 3, 7, and 14, corneal fluorescein staining was performed using slit-lamp biomicroscopy. The animals were sacrificed, and a terminal deoxynucleotidyl transferase dUTP nick end labeling (TUNEL) assay and immunofluorescent staining for MT2 receptors, light chain 3-II (LC3-II), p62, and BAX were performed. In Part II, the tear volume and TBUT were measured on days 7 and 14. After euthanasia on day 14, 2′,7′-dichlorodihydrofluorescein diacetate (DCF-DA) staining and flow cytometry were performed. Experiments were conducted using three independent sets of mice (30 mice per set in Part I and twelve mice per set in Part II).

### 4.2. Immunofluorescent Staining

Immunofluorescence staining was performed on cryosections of the eye and cornea. Sections were fixed in acetone at −20 °C and then incubated overnight at 4 °C with mouse monoclonal MT2 antibody (1:50, catalog no. OABF00337, Aviva, CA, USA), LC3-II (1:50; catalog no. ab192890; Abcam, Cambridge, UK), p62 (1:50; catalog no. ab211324), and BAX antibodies (catalog no. ab216494). The next day, the samples were incubated with Alexa Fluor488-conjugated chicken anti-mouse IgG (1:200; catalog no. A21200; Invitrogen, Eugene, OR, USA) for 1 h in the dark at room temperature, which was followed by three washes with PBS. Thereafter, sections were counterstained with 4′,6-diamidino-2-phenylindole (DAPI; Catalog No. H-1200; Vector, Burlingame, CA, USA) for 5 min. Digital images of representative areas of the conjunctiva and lacrimal glands were captured using a Leica upright microscope (DM2500; Leica Microsystems, Wetzlar, Germany). The results were expressed as the intensity of the expression signal, which was quantified as the mean fluorescence intensity (MFI) from three sections per eye.

### 4.3. TUNEL Assay

A TUNEL assay was used to detect the 3ʹ-hydroxyl ends of fragmented DNA to identify apoptotic cells in corneal tissues. Staining was evaluated using the DeadEnd Fluorometric TUNEL System (Promega, Madison, WI, USA). Cell images were obtained separately at excitation wavelengths of 405 nm and 488 nm and emission wavelengths of 424–472 nm and 502–50 nm for the TUNEL assay and DAPI staining, respectively. TUNEL-positive cells and the DAPI staining of cell nuclei were observed under a microscope at 20× magnification.

### 4.4. Evaluation of Ocular Surface Parameters

Tear volume was measured using phenol red cotton threads (Zone-Quick; Oasis, Glendora, CA, USA) as previously described [36]. The threads were applied to the lateral canthus for 20 s. Tear volume, expressed as the length of the thread that turned due to tear fluid, was measured using a microscope (SMZ 1500; Nikon, Melville, NY, USA). A standard curve was generated to convert the distance into volume [36].

Next, 1% sodium fluorescein (1 μL) was instilled into the inferior conjunctival sac. After three blinks, the TBUT was recorded in seconds under a slit-lamp biomicroscope (BQ-900; Haag-Streit, Bern, Switzerland) with cobalt blue light.

After 90 s, a researcher who was blinded to the therapeutic conditions evaluated the punctate staining of the cornea. Each cornea was divided into four quadrants for the CFS. The CFS was assessed using a 4-point scale (0–4) based on a previous study as follows: 0, absent; 1, slight punctate staining, <30 spots; 2, punctate staining > 30 spots, but non-diffuse; 3, diffuse staining, without a positive plaque; and 4, positive fluorescein plaque. The scores for the four quadrants were summed to generate a final score (maximum score of 16 points) [37].

### 4.5. DCF-DA for ROS Production

Levels of extracellular reactive oxygen species (ROS) were measured using a CM-H2DCFDA kit according to the manufacturer’s protocol, as previously described [4]. In brief, the cornea and conjunctiva of mice were surgically harvested 14 days post-treatment; then, the cells were centrifuged at 450× *g* for 7 min at 4 °C, washed with PBS and 10 μM DCF-DA, and incubated for 30 min at 37 °C. The cells were analyzed using the FACSCalibur flow cytometer (BD Biosciences, San Jose, CA, USA) at an excitation wavelength of 480 nm and an emission wavelength of 530 nm. Results were expressed as the mean percentage increase in DCF-DA fluorescence relative to the normal control tissue using Cell Quest software (version 5.2.1; BD Biosciences).

### 4.6. Flow Cytometry

Flow cytometry was performed to determine the number of CD4^+^ interferon (IFN)-γ^+^ T cells in the cornea and conjunctiva using a previously described method [38]. The samples were incubated with fluorescein-conjugated anti-CD4 antibody (BD Biosciences), phycoerythrin-conjugated anti-IFN-γ antibody (BD Biosciences), and isotype control antibody at 37 °C for 30 min. The number of CD4^+^ IFN-γ^+^ T cells was counted using a FACSCalibur cytometer running Cell Quest software (version 5.2.1; BD Biosciences).

### 4.7. Statistical Evaluation

Data were statistically analyzed using SPSS (version 18.0; Chicago, IL, USA). Results were expressed as mean ± standard deviation. Statistical analysis employed one-way ANOVA followed by Tukey’s post hoc test. All comparisons were subjected to a statistical significance test with *p* < 0.05.

## 5. Conclusions

In summary, this study provides evidence that the selective MT2 receptor agonist IIK7 offers significant protective effects against BL-induced corneal damage. By modulating autophagy, reducing apoptosis, and mitigating ROS, IIK7 helps preserve the integrity and function of the corneal epithelium induced by BL exposure. The findings suggest that targeting MT2 receptors with selective agonists like IIK7 may be a promising strategy for managing BL-induced ocular surface disorders and corneal stress response mechanisms. This study contributes to developing therapies to address the adverse effects of increased BL exposure in the modern digital age.

## Figures and Tables

**Figure 1 ijms-25-11243-f001:**
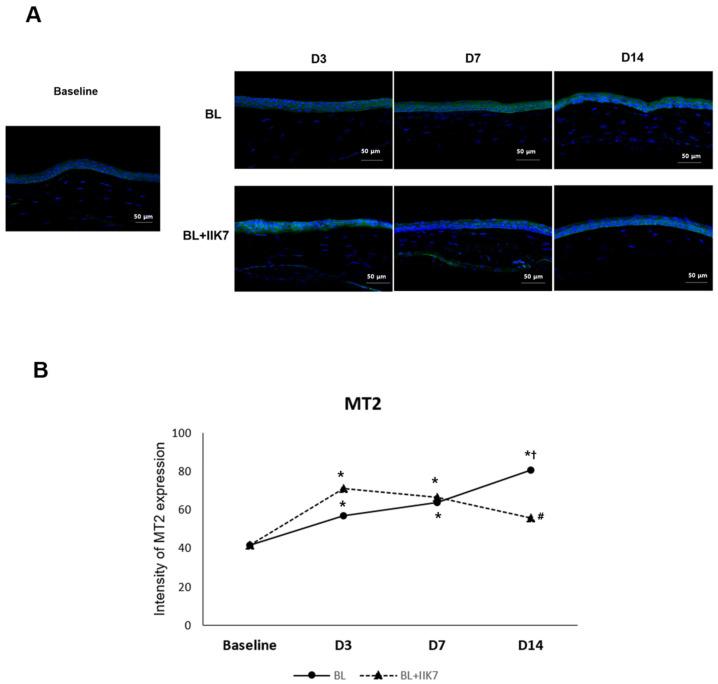
Corneal MT2 receptor expression in BL-exposed (BL) and BL-exposed with IIK7 treatment (BL + IIK7) groups at days 3, 7, and 14. Representative immunofluorescence images (**A**) and quantitative analysis of MT2 receptor expression levels (**B**). Data are presented as mean ± SD. * *p* < 0.05 vs. baseline, ^†^
*p* < 0.05 vs. day 3, # *p* < 0.05, BL + IIK7 vs. BL.

**Figure 2 ijms-25-11243-f002:**
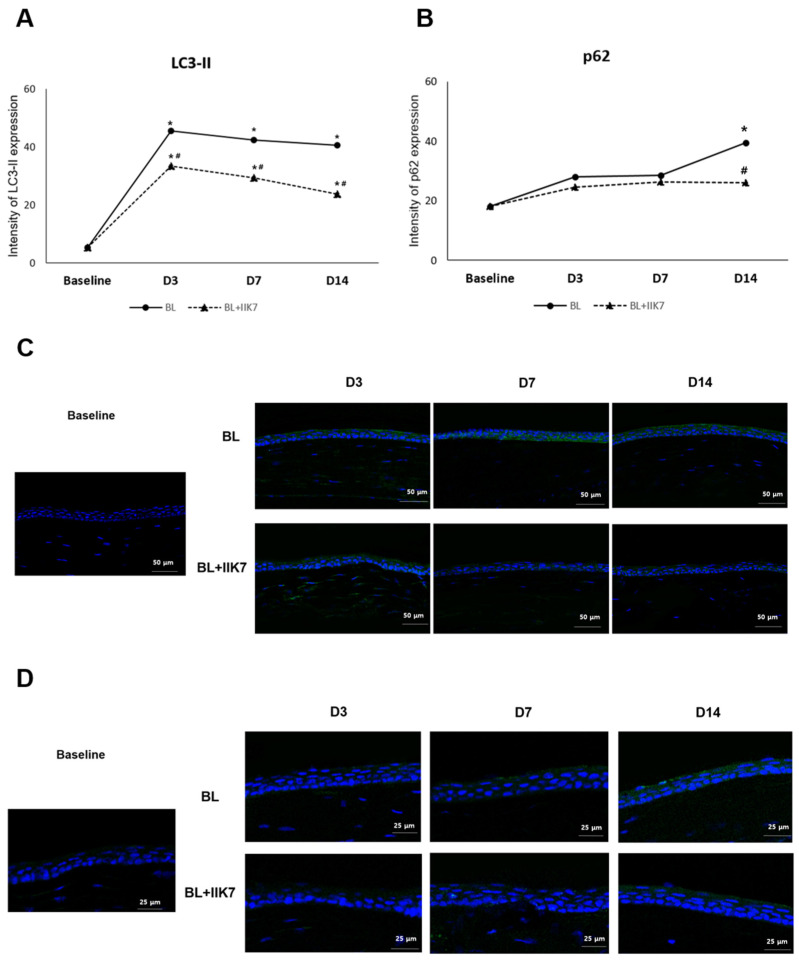
Levels of immunofluorescence of autophagy markers, LC3-II (**A**) and p62 (**B**) in the corneal epithelium on days 3, 7, and 14 in BL-exposed (BL) and BL-exposed with IIK7 treatment (BL + IIK7) groups. * *p* < 0.05 vs. baseline, # *p* < 0.05, BL + IIK7 vs. BL. Representative immunofluorescence of LC3-II (**C**) and p62 (**D**).

**Figure 3 ijms-25-11243-f003:**
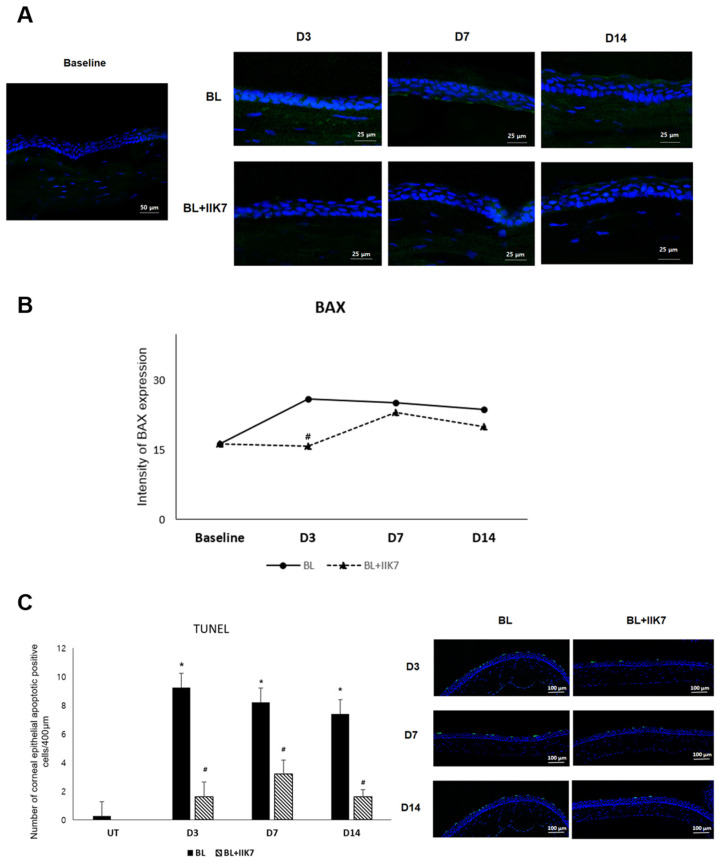
Representative immunofluorescence images of apoptotic marker, BAX (**A**), and intensity of immunofluorescence (**B**) on days 3, 7, and 14 in BL-exposed (BL) and BL-exposed with IIK7 treatment (BL + IIK7) groups. The mean number of the TUNEL-positive cells and its representative images (**C**). * *p* < 0.05 vs. baseline, # *p* < 0.05, BL + IIK7 vs. BL.

**Figure 4 ijms-25-11243-f004:**
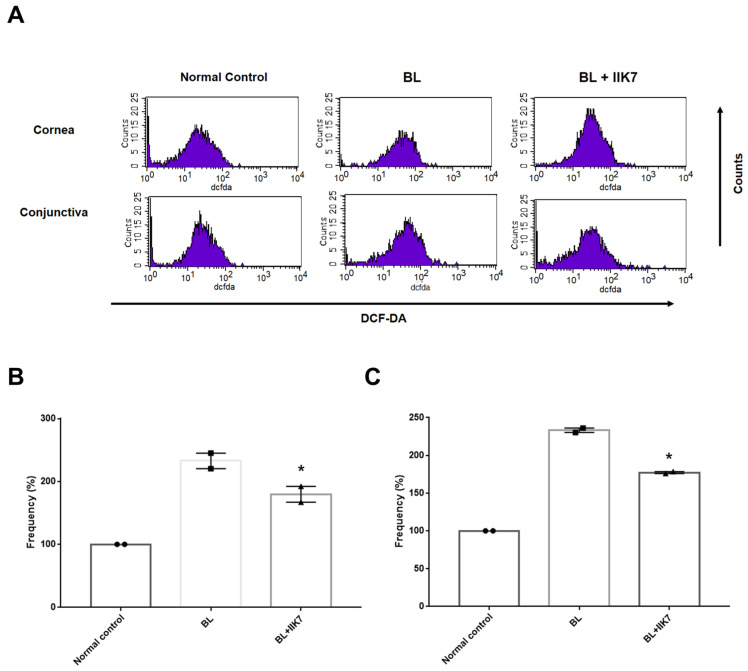
Levels of extracellular ROS measured using 2′,7′-DCF-DA in representative images (**A**), and frequency of DCF-DA in the cornea (**B**), conjunctiva (**C**) of the normal control, BL-exposed (BL), and BL-exposed with IIK7 treatment (BL + IIK7) group at day 14. * *p* < 0.05, BL + IIK7 vs. BL.

**Figure 5 ijms-25-11243-f005:**
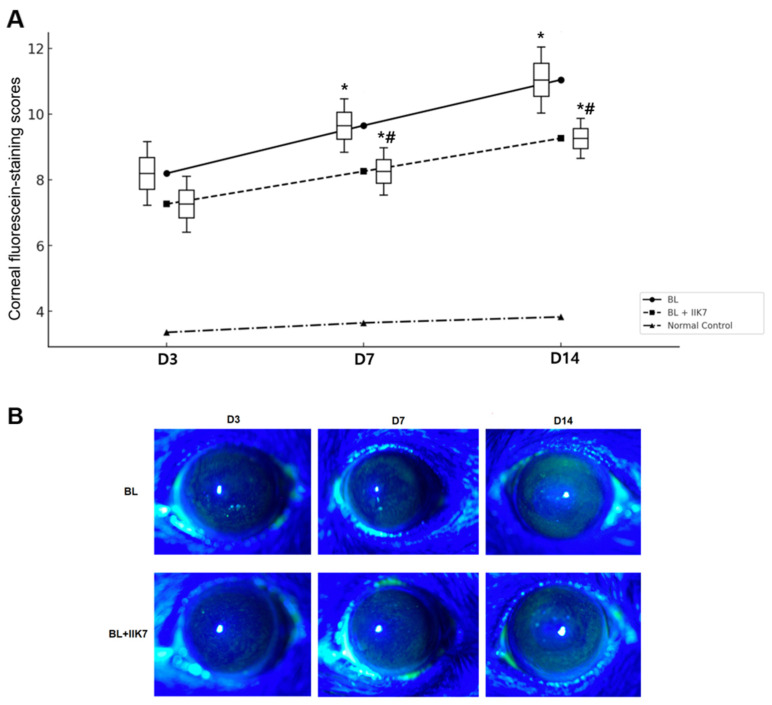
Mean corneal fluorescein-staining scores (**A**) and representative photographs (**B**) of BL-exposed (BL) and BL-exposed with IIK7 treatment (BL + IIK7) groups at days 3, 7, and 14. * *p* < 0.05 vs. normal control, # *p* < 0.05, BL + IIK7 vs. BL.

**Figure 6 ijms-25-11243-f006:**
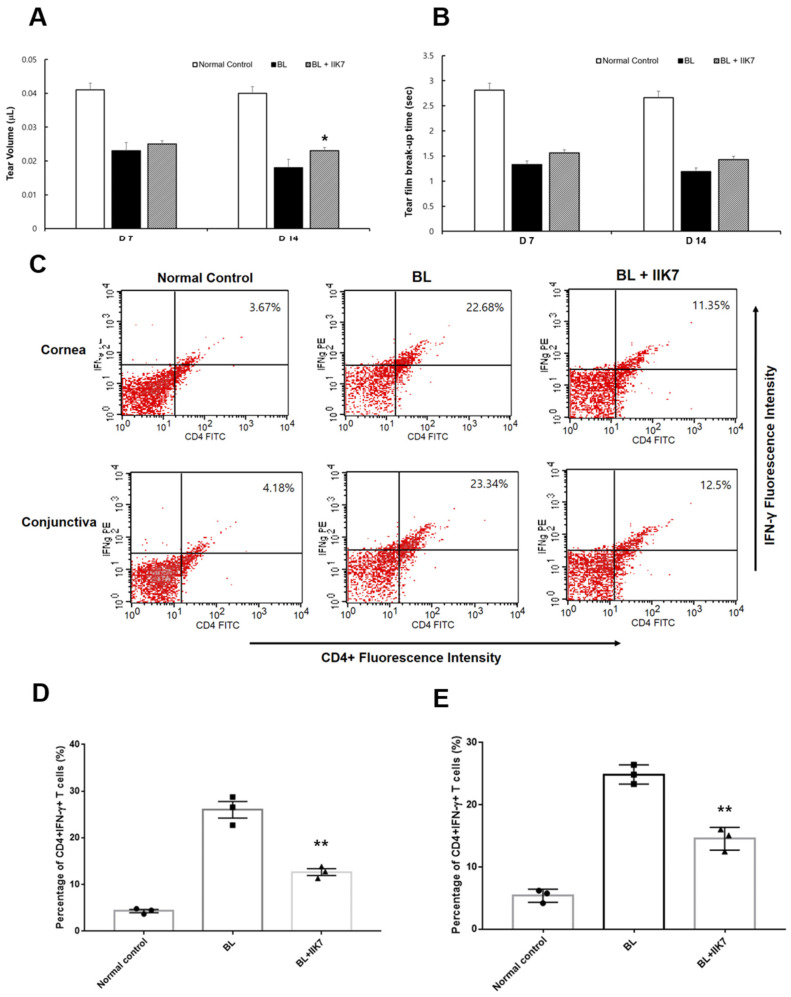
Mean tear volumes (**A**), tear-film break-up time (**B**) of BL-exposed (BL) and BL-exposed with IIK7 treatment (BL + IIK7) groups at days 7 and 14. Flow cytometry showing CD4^+^ IFN-γ^+^ T cells in representative images (**C**) and the mean percentages of CD4^+^ IFN-γ^+^ T cells in the cornea (**D**) and conjunctiva (**E**). * *p* < 0.05 vs. BL group, ** *p* < 0.01 vs. BL group.

## Data Availability

The datasets used and/or analyzed during the current study available from the corresponding author on reasonable request.

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
