# Peer review of "A Selective Melatonin 2 Receptor Agonist, IIK7, Relieves Blue Light-Induced Corneal Damage by Modulating the Process of Autophagy and Apoptosis"

_ijms, 2024, doi:10.3390/ijms252011243_

Round 1
Reviewer 1 Report
Comments and Suggestions for Authors
This study investigated the effect of the selective MT2 receptor agonist, IIK7, on corneal autophagy and apoptosis in mice, aiming to demonstrate the role of MT2 receptor in reducing corneal epithelial damage and inflammation from blue light exposure in mice. There are some questions should be clarified.
1. light induction performed twice dailyà how long per light induction?
2. The anti-inflammation or reduce ROS from blue light exposure in mice cornea by MT2 receptor agonist require more evidence to support. It should provide more solid findings from your or others study in the discussion.
Author Response
1. Light induction performed twice dailyà how long per light induction?
: Thank you for your comment. Each light induction was performed for 74 minutes, twice daily.
|
Page 10, line 263: mice were exposed to BL at an energy dose of 100 J/cm2 (74 min) at 3, 7, or 14 days, ~ |
- The anti-inflammation or reduce ROS from blue light exposure in mice cornea by MT2 receptor agonist require more evidence to support. It should provide more solid findings from your or others study in the discussion.
: Thank you for your valuable comment. We will enhance the discussion section as follows.
|
- Page 10, line 232-237: Mitochondrial dysfunction, a key trigger of the intrinsic apoptosis pathway, was asso-ciated with increased ROS production [33]. By stabilizing mitochondrial function and lowering ROS levels, IIK7 mitigates oxidative stress, reducing both apoptosis and in-flammation [33,34]. Moreover, by decreasing ROS, IIK7 indirectly suppresses inflam-mation by limiting the activation of inflammatory pathways like NF-κB and reducing pro-inflammatory cytokine production [32,34,35]. |

Reviewer 2 Report
Comments and Suggestions for Authors
The current manuscript aims to describe that a selective melatonin 2 receptor agonist (IIK7) relieves blue light-induced corneal damage by modulating the process of autophagy and apoptosis. Although the topic is interesting in its scientific field, there are some issues that require the authors’ attention to improve the quality of this particular manuscript before further consideration for publication in a high-quality journal “IJMS”.
Specific comments:
1. According to the data presentation in Figure 1B, the MT2 expression in the BL group is increased with time. But, the underlying reason for this observation is unclear to the readers. Please justify.
2. This work focuses on IIK7. But, whether other agonists of melatonin receptors such as MT1 and MT3 can exhibit similar protective effects against blue light-induced damage? Please clarify.
3. In order to determine the optimal dose of IIK7 for better protection, the experiments should be carried out by using different concentrations of IIK7. Please improve.
4. What are the effects of long-term treatment with IIK7 on cornea in mice exposed to blue light over 14 days? Please justify.
5. What are the downstream signaling pathways involved in IIK7-mediated reduction of reactive oxygen species and regulation of apoptosis? Please specify.
6. As stated by the authors, a TUNEL assay was used to detect the 3ʹ-hydroxyl ends of fragmented DNA to identify apoptotic cells in corneal tissues. Nevertheless, this important experimental claim was not supported by appropriate documentation. If possible, please consider the inclusion of the following case study (DOI: 10.1002/advs.202302174) in the reference list to strengthen the manuscript quality.
Author Response
1. According to the data presentation in Figure 1B, the MT2 expression in the BL group is increased with time. But, the underlying reason for this observation is unclear to the readers. Please justify.
: Thank you for your comment. In our previous study (Jin et al., 2022), we observed a time-dependent increase in MT2 expression in response to blue light (BL) exposure. Specifically, when the MT2 receptor was inhibited by an antagonist, we observed increased impaired autophagy and enhanced apoptosis, leading to aggravate corneal epithelial damage. The increase in MT2 expression over time likely reflects the progressive need for cellular protection against the sustained oxidative stress caused by prolonged BL exposure.
|
Page 9, lines 180-184: In our previous studies, we observed a time-dependent increase in MT2 expression in response to BL exposure [15]. Specifically, inhibiting MT2 with antagonist led to increased impaired autophagy and apoptosis, which aggravated corneal epithelial damage [15]. The increasement in MT2 expression over time reflects the need for cellular protection against the sustained oxidative stress caused by prolonged BL exposure. |
2. This work focuses on IIK7. But, whether other agonists of melatonin receptors such as MT1 and MT3 can exhibit similar protective effects against blue light-induced damage? Please clarify.
: Thank you for your comment. In our previous research (Jin et al., 2022), we specifically observed that MT1 expression in the cornea did not significantly increase in response to BL exposure, whereas MT2 expression was significantly upregulated. This suggests that MT2 is the more relevant receptor involved in mitigating BL-induced damage. Furthermore, there is no direct evidence in the literature supporting the expression of MT3 in the cornea. Therefore, MT2 serves as a more appropriate target for therapeutic intervention in BL-induced damage, which is why we focused on IIK7, a selective MT2 agonist, in this study.
|
Page 9, lines 184-188: MT2 appears to play a central role in mitigating BL-induced damage, as MT1 did not exhibit upregulation in response to BL exposure, and current literature provides no evidence for MT3 expression in the cornea [13,15]. Therefore, we focused on a selective MT2 agonist as the primary candidate for therapeutic intervention in this study. |
3. In order to determine the optimal dose of IIK7 for better protection, the experiments should be carried out by using different concentrations of IIK7. Please improve.
: Thank you for your comment. We conducted a preliminary study using IIK7 at concentrations of 0.1, 1, and 10 nM. Based on the results shown in the table below, we observed that 1 nM provided the optimal balance of efficacy and safety, which is why we selected this concentration for the main study. (We attached additional table.)
4. What are the effects of long-term treatment with IIK7 on cornea in mice exposed to blue light over 14 days? Please justify.
: Thank you for your comment. Our study focused on investigating the acute reactions to BL exposure, which is why we limited the experiment to 14 days. Extending the exposure period beyond this time frame could result in more complex and less predictable physiological responses, making it difficult to isolate the effects of IIK7. However, we agree that longer exposure periods might yield different outcomes, and we will address this as a limitation in the discussion of our manuscript.
|
Page 10, lines 250-252: The limitation of our study is that we focused on the acute effects of blue light exposure over a 14-day period. Extending exposure could lead to more complex and unpredictable physiological responses. |
5. What are the downstream signaling pathways involved in IIK7-mediated reduction of reactive oxygen species and regulation of apoptosis? Please specify.
Thank you for your comment. The apoptosis observed in our study is primarily associated with the intrinsic (mitochondrial) pathway. This pathway is regulated by BAX, a pro-apoptotic protein, which promotes mitochondrial outer membrane permeabilization, leading to the release of cytochrome c and the activation of caspase-9 and caspase-3. In both this and our previous studies, we observed that BL exposure upregulated both BAX and cleaved caspase-3, indicating the activation of the intrinsic pathway. IIK7, through MT2 receptor activation, may potentially modulate apoptosis by interrupting the intrinsic apoptosis pathway.
|
Page 9, lines 216-222: The apoptosis observed in our study is primarily associated with the intrinsic (mitochondrial) pathway [29,30]. This pathway is regulated by BAX, a pro-apoptotic protein, which promotes mitochondrial outer membrane permeabilization, leading to the release of cytochrome c and the activation of caspase-9 and caspase-3 [29,30]. In both this and our previous studies, we observed that BL exposure upregulated both BAX and cleaved caspase-3, indicating the activation of the intrinsic pathway [15]. |
- As stated by the authors, a TUNEL assay was used to detect the 3ʹ-hydroxyl ends of fragmented DNA to identify apoptotic cells in corneal tissues. Nevertheless, this important experimental claim was not supported by appropriate documentation. If possible, please consider the inclusion of the following case study (DOI: 10.1002/advs.202302174) in the reference list to strengthen the manuscript quality.
à Thank you for the suggestion. We agree that including the mentioned case study would strengthen our manuscript. As recommended, we will add this reference to the manuscript's reference list.
|
32. Yang, C.-J.; Nguyen, D.D.; Lai, J.-Y. Poly(l-Histidine)-Mediated On-Demand Therapeutic Delivery of Roughened Ceria Nanocages for Treatment of Chemical Eye Injury. Adv Sci (Weinh) 2023, 10, e2302174, doi:10.1002/advs.202302174. |

Round 2
Reviewer 1 Report
Comments and Suggestions for Authors
no more comment
Reviewer 2 Report
Comments and Suggestions for Authors
The revised version has adequately addressed most of the critiques raised by this reviewer and is now suitable for publication in "IJMS".